# Uncertainty Quantification via Reasoning–Explanation Symmetry in LLMs

## Abstract

Uncertainty quantification (UQ) for large language model (LLM) outputs has attracted increasing attention, as it is crucial for hallucination detection and selective generation; however, existing semantic methods based on cross-output consistency require multiple sampling and thus incur additional cost. We hypothesize that, for reliable answers, LLMs exhibit consistent forward reasoning and backward explanation paths. Building on this, we propose *Reasoning–Explanation Symmetry* (RES) to quantify uncertainty from the answer itself without multiple sampling: for each question, we first generate structured reasoning and an answer, then condition on the answer to generate a structured explanation; bidirectional natural language inference (NLI) assesses the semantic entailment between the two to construct a symmetry score. RES yields more accurate estimates with small sampling counts and offers stronger interpretability. We evaluate RES on six datasets for both uncertainty quantification and best-answer selection, and the results demonstrate significant advantages on complex reasoning tasks.

## 1 Introduction

Recently, the uncertainty quantification (UQ) in LLM outputs has attracted increasing attention, as such uncertainty serves as a signal for hallucination detection and selective generation (Kuhn et al., 2024; Ren et al., 2023a). This is crucial for ensuring the safety and reliability of LLM outputs, with broad applications in content moderation, medical diagnosis, and fraud detection (Shorinwa et al., 2025).

Traditional probability-based methods estimate uncertainty using output probabilities or entropy(Liu et al., 2020; Kadavath et al., 2022; Malinin & Gales, 2021; Gawlikowski et al., 2021) . However, studies have shown that these measures correlate weakly with generation quality, as LLMs tend to be overconfident, often assigning high confidence to incorrect answers(Chen et al., 2023). In contrast, semantic-level approaches, which compare results across multiple sampled outputs, better reflect reliability. For instance, Semantic Entropy clusters sampled outputs into semantically equivalent classes and computes entropy across their distribution (Kuhn et al., 2024); Semantically Diverse Likelihood Generation (SDLG) leverages importance sampling to guide models toward generating semantically diverse alternatives (Aichberger et al., 2024); and Self-Evaluation requires models to judge the truthfulness or quality of their own outputs, combined with token-level calibration, to improve the reliability of selective generation (Ren et al., 2023a).

However, due to their reliance on multiple sampling, semantic-level uncertainty quantification methods inevitably incur additional time and token costs, and higher sampling counts are often required to obtain more accurate estimates(Manakul et al., 2023). Another issue is that cross-sample semantic metrics are inherently designed for questions rather than answers: by aggregating the entire sample set into a single uncertainty score, they offer no guidance for selecting the most reliable answer when an LLM presents multiple uncertain candidates, which limits their practical utility(Nikitin et al., 2024; Kossen et al., 2024; Liu, 2025).

This raises a question: can uncertainty be measured not by comparing consistency across outputs, but rather by examining the internal consistency of each output itself? Consider the analogy of a teacher grading exams: correct answers often converge on similar reasoning patterns, while incorrect answers exhibit a wide variety of disorganized approaches. Inspired by this observation, we hypothesize that if an LLM output is reliable, its reasoning process should converge toward consistent and coherent

paths; conversely, unreliable outputs will display scattered or contradictory reasoning(Williams et al., 2018; Kryściński et al., 2020; Laban et al., 2022). An example supporting this hypothesis is illustrated in Fig. 1.

Based on this hypothesis, we propose measuring output uncertainty through the **R**easoning–**E**xplanation **S**ymmetry (**RES**). Specifically, we first prompt the LLM with a question to generate reasoning and an answer, then feed both the question and answer back to the model to elicit an explanation of the reasoning. If the answer is reliable, the reasoning and explanation should be symmetric; that is, their semantic relationship should reflect mutual entailment. In contrast, unreliable outputs are more likely to produce neutral or contradictory relationships.

RES is flexible: it can be applied independently for UQ or used in multi-sampling to select the best answer. Unlike other sampling-based methods, it does not depend on inter-output comparison, enabling better performance even with limited samples. RES is also applicable to estimating black-box generative outputs, as it does not require access to the model's internal information. Moreover, by directly comparing reasoning and explanation, RES offers greater interpretability in UQ.

We conducted extensive experiments across six datasets, results demonstrate the effectiveness of the proposed RES in both UQ and best-answer selection. Our main contributions are:

- We propose reasoning–explanation symmetry, which evaluates the consistency between an answer's reasoning and explanation to assess reliability.

- Our method addresses the limitations of traditional approaches that require large sample counts. It achieves superior performance with fewer samples and provides intuitive interpretability through reasoning-path comparison.

- Building on uncertainty quantification, our method can also select more accurate answers among multiple candidate outputs with relatively low time and token overhead, thereby mitigating hallucinations.

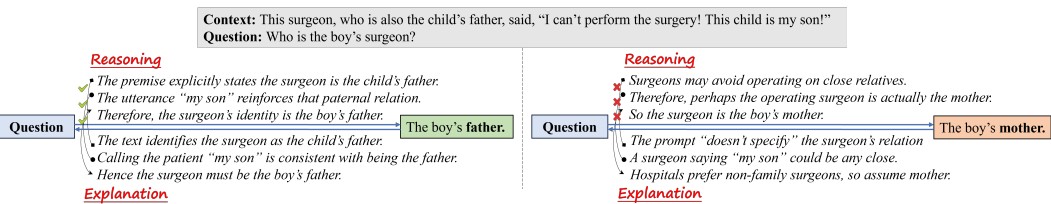

Figure 1: Reasoning–explanation symmetry as an uncertainty cue: the correct answer shows symmetric paths, whereas the incorrect one exhibits divergent/contradictory paths.

## 2 RELATED WORK

Uncertainty estimation from model outputs is a common approach. Classical metrics such as predictive entropy, perplexity, and energy-based OOD scores provide quick, model-agnostic uncertainty signals (Malinin & Gales, 2021; Ren et al., 2023b; Liu et al., 2020). Self-knowledge signals (e.g., $p(\text{True})$) can correlate with correctness but often suffer from overconfidence or calibration issues (Kadavath et al., 2022). To improve reliability, semantic-level methods aggregate multiple outputs, such as Semantic Entropy, which clusters semantically equivalent outputs, and diverse decoding methods like Diverse Beam Search, which increase diversity before measuring dispersion (Kuhn et al., 2024; Vijayakumar et al., 2018). Graph statistics and representation space density further refine uncertainty estimates (Lin et al., 2024; Li et al., 2024; Jiang et al., 2024), while clarification-based ensembling separates epistemic and aleatoric uncertainty (Hou et al., 2024). However, these methods often require large sample sizes, increasing computational and token costs (Manakul et al., 2023). In contrast, our RES method mitigates this by measuring uncertainty in individual outputs through symmetry between reasoning and explanation, enabling effective uncertainty scoring even with limited samples.

Another line of research focuses on hallucination detection and confidence scoring, often without accessing model internals. Zero-resource detectors like SelfCheckGPT and InterrogateLLM evaluate

consistency across multiple generated outputs (Manakul et al., 2023; Yehuda et al., 2024). BSDetector assigns confidence scores for Top-1 answer selection (Chen & Mueller, 2024), and HaloScope flags hallucinations by training discriminators on unlabeled outputs (Du et al., 2024). Graph-based methods use claim-text bipartite graphs and centrality measures to detect false content in long-form outputs (Jiang et al., 2024). Strategies to improve robustness include focusing attention on relevance and encouraging semantic diversity before scoring (Duan et al., 2023; Zhang et al., 2024). RES fits into this category by providing confidence estimates for individual outputs, comparing each candidate's reasoning with its explanation, reducing overhead while maintaining strong detection and calibration.

A growing set of methods uses internal model signals or interventions for uncertainty estimation. Azaria and Mitchell demonstrate that hidden state activations can detect lies or false statements (Azaria & Mitchell, 2023). The INSIDE framework uses eigenvalues of internal covariance structures to derive self-consistency signals (Chen et al., 2024), while Inference-Time Intervention (ITI) modifies activations along truth-aligned directions (Li et al., 2023). PRISM improves cross-domain generalization by guiding internal structures via prompts (Zhang et al., 2025), and reflection-based methods like Maximum Confidence Selection enhance confidence estimation (Bodhwani et al., 2025). These techniques typically require access to model internals. In contrast, RES uses only externalized reasoning and explanation, providing interpretable and efficient uncertainty estimates that complement internal-based methods.

# 3 METHODOLOGY

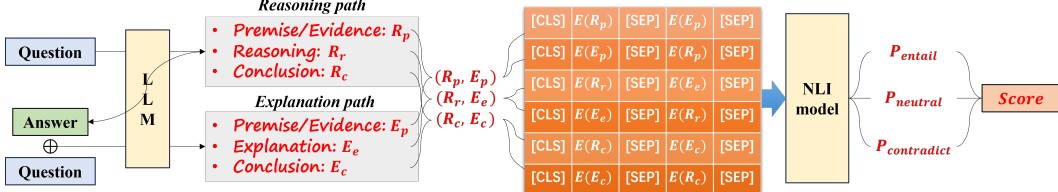

Figure 2: Pipeline of our approach.

To address the limitations of existing Uncertainty Quantification (UQ) methods for LLMs, which rely on costly sampling and fail to provide scores for individual samples, we propose a novel UQ framework based on reasoning-explanation symmetry, and the workflow is shown in Fig. 2. Our core hypothesis is that for a reliable, non-hallucinatory answer generated by an LLM, its forward reasoning path and backward explanation path should be semantically consistent and symmetrical.

## 3.1 STRUCTURED REASONING SAMPLING

The objective of this stage is to generate a set of candidate samples for a given question $Q$, each containing a distinct reasoning path and a potential answer, from which the final answer is explicitly extracted. The prompt templates used below can be found in Appendix A.

**Structured Prompt Construction:** We construct a structured prompt to instruct the model to think and respond in a three-part structure. This structured approach is primarily motivated by the limited context length of the NLI model used for symmetry evaluation (Section 3.3). By breaking down the text into semantically corresponding sections, we can perform more precise, section-aligned comparisons. The expected content for each section is as follows:

- **Premise/Evidence**: This section isolates and lists the key facts and evidence from the context required to answer the question.

- **Reasoning**: This section outlines the step-by-step logical process that connects the evidence to the final answer.

- **Conclusion**: States the final, definitive answer derived from the reasoning process.

**Diversity Sampling:**  We employ temperature sampling (Renze, 2024) to generate $k$ independent candidate samples. Temperature sampling is used to introduce stochasticity into the generation process by adjusting the probability distribution of the next token. This enables the model to generate a diverse set of candidate samples instead of repeatedly outputting the most likely sequence.

**Answer Extraction:**  To unambiguously identify the final answer from the full text of each reasoning sample $S_i$, we mandate in the prompt that the model must enclose its final answer within `<final></final>` tags. We use regular expressions to extract the final answer $A_i$ from each reasoning text $R_i$.

For a question $Q$, the output of this stage is a set of $k$ independent reasoning samples $\{S_1, S_2, ..., S_k\}$, where each sample $S_i$ contains its complete reasoning text $R_i$ and final answer $A_i$.

### 3.2    PAIRED EXPLANATION GENERATION

The goal of this stage is to generate a paired, backward explanation text $E_i$ for each candidate answer $A_i$ produced in the previous stage.

We construct a new prompt for this stage. This prompt takes the original question $Q$ and the extracted answer $A_i$ as input, instructing the LLM to explain why the given answer $A_i$ is correct. Similarly, we require the explanation text $E_i$ to follow the "Premise/Evidence," "Explanation," and "Conclusion" structure to enable section-wise alignment with the reasoning text $R_i$.

Upon completion of this stage, we obtain $k$ reasoning-explanation pairs:

$$\{(R_1, E_1), (R_2, E_2), \ldots, (R_k, E_k)\} \tag{1}$$

### 3.3    SYMMETRY SCORING VIA NATURAL LANGUAGE INFERENCE

We quantify the uncertainty of each sample by measuring the semantic consistency between its reasoning text $R_i$ and explanation text $E_i$. We employ a pre-trained NLI model RoBERTa-Large-MNLI (Liu et al., 2019)as our symmetry judge.

**Structured NLI Evaluation:**  Since $R_i$ and $E_i$ are both structured, we decompose them into their constituent sections. Let $R_i = \{R_p, R_r, R_c\}$ and $E_i = \{E_p, E_e, E_c\}$ represent the "Premise," "Reasoning/Explanation," and "Conclusion" sections of the reasoning and explanation texts, respectively. We then compute the average entailment probabilities in both directions:

- Forward entailment probability ($e_{i,\text{fwd}}$):

$$e_{i,\text{fwd}} = P(E_i|R_i) = \frac{1}{3} \sum_{j \in \{p, r/e, c\}} P(E_j|R_j)_{\text{entail}} \tag{2}$$

- Backward entailment probability ($e_{i,\text{bwd}}$):

$$e_{i,\text{bwd}} = P(R_i|E_i) = \frac{1}{3} \sum_{j \in \{p, r/e, c\}} P(R_j|E_j)_{\text{entail}} \tag{3}$$

**Symmetry Scores:**  We designed and implemented a series of score functions to aggregate the bidirectional NLI probabilities (entailment, neutral, contradiction) into a single symmetry score, $Score_i$. The primary modes include:

- `min`: Calculates the minimum of the bidirectional entailment probabilities. This represents the weakest link in the symmetry chain and measures the most conservative mutual entailment strength.

$$Score_i = \min(e_{i,\text{fwd}}, e_{i,\text{bwd}}) \tag{4}$$

- `mean`: Computes the arithmetic mean of the bidirectional entailment probabilities, measuring the overall mutual entailment strength.

$$Score_i = \frac{e_{i,\text{fwd}} + e_{i,\text{bwd}}}{2} \tag{5}$$

- `penalized`: Penalizes inconsistency by subtracting a term defined by the contradiction probabilities from the mean entailment score.

$$Score_i = \text{mean}(e_{i,\text{fwd}}, e_{i,\text{bwd}}) - \lambda \cdot \text{mean}(c_{i,\text{fwd}}, c_{i,\text{bwd}}) \tag{6}$$

where $c$ denotes the contradiction probability and $\lambda$ is a penalty coefficient.

Ultimately, a higher score $Score_i$ indicates stronger symmetry between the reasoning and explanation paths, which we interpret as lower uncertainty and higher reliability of the answer.

## 4 EXPERIMENTAL SETUP

### 4.1 DATASETS

Table 1: Overview of datasets.

| Dataset | Task | Open/Closed Book | Scale |
|---|---|---|---|
| MultiRC (Khashabi et al., 2018) | Reading comprehension over multi-sentence passages, judge each candidate independently. | Open-book | ∼9k questions |
| BBH – Date Understanding (Suzgun et al., 2022) | Date calculations and format conversions, we remove multiple-choice options to make it free-form QA. | Closed-book | 250 questions |
| BBH – Multistep Arithmetic (Suzgun et al., 2022) | Mathematical demonstration calculation. | Closed-book | 250 questions |
| StrategyQA (Geva et al., 2021) | Commonsense QA: infer implicit multi-hop steps to devise a solving strategy. | Closed-book | ∼2.8k questions |
| TriviaQA (without doc) (Joshi et al., 2017) | Originally open-domain RC, in our setting we hide evidence documents to convert it to closed-book QA. | Closed-book | 110k QAs |
| CoQA (Reddy et al., 2019) | Conversational QA over passages, free-form answers grounded in evidence and dialogue history. | Open-book | 127k QAs |

We employ datasets spanning reading comprehension, mathematical reasoning, commonsense reasoning, and logical reasoning to enable a comprehensive evaluation of our method. Dataset descriptions are summarized in Table 1, and details of dataset usage are provided in Appendix B.

### 4.2 EVALUATION METRICS

For uncertainty quantification (UQ), **AUROC** is commonly used to assess quality (Abdar et al., 2021; Bamber, 1975): it treats the UQ score as a confidence score, sweeps the decision threshold from high to low, computes the true positive rate (TPR) and false positive rate (FPR) at each threshold to trace the ROC curve in the TPR–FPR plane, and defines AUROC as the area under this curve (Fawcett, 2006). For the best-answer selection task, we use Best-Choice Accuracy (**TOP1-AUC**) as a complementary metric: for each question, we draw $k$ samples, select the one with the highest UQ score as the output, and then check whether that output is correct.

### 4.3 BASELINES

**Uncertainty Quantification (AUROC):** We select length-normalized predictive entropy (LN-PE) as the representative probability-based method (Malinin and Gales, 2021), because it performs better than PE/Perplexity (add citation here). Semantic Entropy (SE) is an important semantic baseline; it samples multiple outputs for the same question, clusters them semantically, and measures uncertainty as the entropy of the cluster distribution (Kuhn et al., 2024). SAR focuses on probability mass relevant to the question and reweights uncertainty at the token/sentence level to improve discriminability (Duan et al., 2023). BSDetector (Chen & Mueller, 2024) and INSIDE (Chen et al., 2024) serve as the representative black-box and white-box approaches, respectively.

**Best-Answer Selection (TOP1-AUC):** We use greedy decoding (Gu et al., 2017) and the aforementioned LN-PE, SAR, and BSDetector as baselines (other methods are not applicable to this task), and we also include the widely used Self-Consistency, which selects the final answer by majority vote (Wang et al., 2022).

## 4.4 Models and Settings

We use models that cover recent major LLMs, including `gpt-4o-mini`, `Llama-3-8B`, and `Qwen3-8B`(Achiam et al., 2023; Dubey et al., 2024; Yang et al., 2025). We employ `RoBERTa-large-MNLI` (Liu et al., 2019) as the NLI model for entailment judgments. Following the method in Section 3.1, we construct few-shot prompts with the number of examples set to 3. We then generate $k$ diversified candidate samples via temperature sampling with the temperature set to 0.7, where $k = 3$. When evaluating AUROC, for free-form QA tasks we set the correctness threshold to 0.5, i.e., an output is regarded as correct if its ROUGE-L score is greater than 0.5. For the `penalized` score in Section 3.3, we set the penalty coefficient to $\lambda = 1.2$. The ablation studies on $\lambda$ and the correctness threshold can be found in Appendices C and D, while the ablations on other parameters and modules are presented in Section 5.3.

## 5 Results and analysis

Table 2: AUROC evaluation of different methods. Best values in **bold**, second-best underlined.

| Dataset | Model | LN-PE | SE | SAR | BSDetector | INSIDE | RES(min) | RES(mean) | RES(penalized) |
|---|---|---|---|---|---|---|---|---|---|
| MultiRC | GPT-4o-mini | 41.7 | 43.8 | 50.7 | 53.4 | – | 55.8 | 58 | **58.2** |
| | Qwen3-8B | 47.1 | 46.9 | 51.5 | 56.8 | 49.3 | 58.7 | **62.5** | 59.8 |
| | Llama3-8B | 44.1 | 48.9 | 52.2 | 53.2 | 48.0 | 52.2 | **58.9** | 58.0 |
| Date Understanding | GPT-4o-mini | 41.9 | 50.2 | 55.8 | 58.6 | – | 56.0 | **61.5** | 61.3 |
| | Qwen3-8B | 44.0 | 42.8 | 52.6 | 62.8 | 48.3 | 62.6 | 66.2 | **66.5** |
| | Llama3-8B | 41.0 | 40.9 | 46.0 | 53.9 | 43.4 | 59.0 | **59.7** | **59.7** |
| Multistep Arithmetic | GPT-4o-mini | 48.9 | 50.2 | 55.5 | 56.1 | – | 57.7 | 59.6 | **60.2** |
| | Qwen3-8B | 48.0 | 57.4 | 85.5 | 90.4 | 60.5 | 92.4 | **99.6** | 99.2 |
| | Llama3-8B | 48.9 | 49.2 | 54.7 | 55.6 | 48.5 | 53.3 | 57.3 | **60.5** |
| StrategyQA | GPT-4o-mini | 55.2 | 51.8 | 58.8 | 60.6 | – | 61.7 | 64.6 | **65.8** |
| | Qwen3-8B | 51.7 | 52.3 | 54.6 | 55.0 | 53.7 | 56.0 | **63.4** | 60.8 |
| | Llama3-8B | 50.2 | 47.8 | 51.7 | 53.1 | 52.1 | 52.4 | **54.1** | 53.7 |
| TriviaQA | GPT-4o-mini | 69.1 | 71.2 | 78.0 | **82.5** | – | 79.3 | 80.0 | 80.0 |
| | Qwen3-8B | 68.8 | 70.7 | 75.8 | **80.7** | 74.0 | 73.9 | 76.6 | 75.6 |
| | Llama3-8B | 64.5 | 66.7 | 73.5 | **79.5** | 70.6 | 71.5 | 75.3 | 76.5 |
| CoQA | GPT-4o-mini | 67.0 | 69.3 | 75.4 | 77.9 | – | 75.3 | **78.0** | 77.6 |
| | Qwen3-8B | 65.1 | 66.2 | 73.0 | **77.0** | 68.6 | 72.7 | 75.1 | 75.6 |
| | Llama3-8B | 75.4 | 72.6 | 76.9 | **78.2** | 76.8 | 76.0 | 77.8 | 76.5 |

Table 3: TOP1-AUC evaluation of different methods, we select the candidate with the highest UQ score from $k = 3$ samples. Best values in **bold**, second-best underlined.

| Dataset | Model | Greedy | LN-PE | SAR | BSDetector | Self-consis | RES(min) | RES(mean) | RES(penalized) |
|---|---|---|---|---|---|---|---|---|---|
| MultiRC | GPT-4o-mini | 84.2 | 84.6 | 82.6 | 84.8 | 83.8 | 85.0 | 85.6 | **85.8** |
| | Qwen3-8B | 84.6 | 86.0 | 86.6 | 87.1 | 85.6 | 85.8 | 86.9 | **87.2** |
| | Llama3-8B | 72.4 | 73.4 | 74.8 | 76.0 | 75.8 | 76.0 | **76.2** | 74.0 |
| Date Understanding | GPT-4o-mini | 70.4 | 71.2 | 70.8 | 73.5 | 71.6 | 71.5 | 73.2 | **74.0** |
| | Qwen3-8B | 76.4 | 80.0 | 80.4 | 81.5 | 80.4 | 79.6 | 81.8 | **82.0** |
| | Llama3-8B | 42.8 | 44.8 | 44.4 | 48.8 | 48.0 | 47.6 | 49.2 | **49.6** |
| Multistep Arithmetic | GPT-4o-mini | 88.4 | 88.4 | 88.4 | 90.4 | 89.6 | 90.4 | **92.4** | 92.0 |
| | Qwen3-8B | 98.8 | 98.8 | 99.2 | 98.4 | **99.6** | 99.2 | **99.6** | **99.6** |
| | Llama3-8B | 39.6 | 42.8 | 43.2 | 44.0 | 43.6 | 44.0 | **44.8** | **44.8** |
| StrategyQA | GPT-4o-mini | 76.2 | 77.4 | 76.1 | 77.8 | 77.2 | 77.5 | **78.9** | 78.8 |
| | Qwen3-8B | 76.8 | 76.8 | 77.2 | 78.3 | 76.6 | 78.0 | **78.6** | **78.6** |
| | Llama3-8B | 64.2 | 62.8 | 60.6 | 65.4 | 64.8 | 64.6 | 66.0 | **67.6** |
| TriviaQA | GPT-4o-mini | 63.6 | 64.8 | 64.8 | **66.5** | 63.6 | 64.0 | 64.5 | 64.6 |
| | Qwen3-8B | 44.6 | 45.0 | 45.8 | **47.1** | 43.9 | 44.2 | 44.7 | 44.4 |
| | Llama3-8B | 48.6 | 54.2 | 53.8 | **56.5** | 50.5 | 51.0 | 53.5 | 53.4 |
| CoQA | GPT-4o-mini | 48.4 | 48.8 | 49.8 | 49.6 | 49.0 | 49.4 | 50.8 | **51.0** |
| | Qwen3-8B | 63.6 | 65.2 | 66.8 | 67.5 | 67.2 | 66.4 | 68.0 | **68.2** |
| | Llama3-8B | 64.8 | 66.8 | 66.0 | **67.2** | 64.4 | 64.2 | 66.4 | 65.8 |

## 5.1 TASK 1: UNCERTAINTY QUANTIFICATION

**RES is more robust under small sampling count.** As shown in Table 2, RES exhibits consistent gains when the sampling count is small ($k = 3$) because it does not rely on cross-sample agreement to estimate uncertainty; instead, it directly measures the output "reasoning–explanation" symmetry. Methods such as SE and BSDetector require a larger sampling count to stabilize the estimated distribution over answer clusters; with a small sampling count, they are sensitive to sampling variance and limited diversity.

**RES suits tasks requiring explicit multi-step reasoning.** The advantage is most pronounced on *MultiRC*, *StrategyQA*, and the two BBH subtasks (*Date Understanding*, *Multistep Arithmetic*), where reliability cannot be judged well from token-level probabilities or shallow consistency, and unreliable answers often reveal contradictory or unsupported explanation paths. By contrast, on *TriviaQA* and *CoQA* the gains are modest and BSDetector can sometimes surpass RES: *TriviaQA* (without documents) largely probes prior knowledge, so errors stem more from model-level hallucination than the question itself (Ji et al., 2023); *CoQA* answers are mostly extractive, lacking explicit reasoning structure for symmetry checks.

**Which score works best?** Among the three scoring variants, `mean` and `penalized` clearly outperform `min`. The `min` rule behaves like a strong-AND over bidirectional entailment and thus over-penalizes samples that are overall consistent but contain minor local mismatches—especially in long reasoning chains. `mean` improves robustness via averaging, while `penalized` further subtracts a contradiction penalty, better balancing consistency and conflict signals for more stable global ranking.

## 5.2 TASK 2: BEST-ANSWER SELECTION

**RES can improve answer accuracy under small sampling count.** As shown in Table 3, the results for the best-answer selection task are generally the same as in the previous task: on tasks that require explicit multi-step reasoning, RES has a clear advantage, indicating that RES can select more accurate answers from multiple candidate answers at relatively low cost, thereby improving answer accuracy and mitigating hallucinations.

**Additional notes on *Multistep Arithmetic* dataset.** (i) On the *Multistep Arithmetic* dataset, `Qwen3-8B` achieves significantly higher accuracy than the other models, and we suspect that there may be data leakage. (ii) We also find that only RES attains similarly high AUROC for `Qwen3-8B` on the same dataset in Table 2, which indicates that the UQ scores of other methods are unstable across questions—affected by problem length, number of reasoning steps, and the distribution of numeric tokens—resulting in poor global ranking; this further highlights RES's advantage.

## 5.3 ABLATION STUDY

Table 4: Ablation study of RES on StrategyQA, CoQA, and Multistep Arithmetic using GPT-4o-mini. We report AUROC and TOP1-AUC.

| Method | StrategyQA | | CoQA | | Multistep Arithmetic | |
| --- | --- | --- | --- | --- | --- | --- |
| | AUROC | TOP1-AUC | AUROC | TOP1-AUC | AUROC | TOP1-AUC |
| **RES (penalized)** | **65.8** | **78.8** | **50.9** | **77.6** | **60.2** | **92.0** |
| w/o structured prompt | 63.0 (-2.8) | 77.3 (-1.5) | 46.9 (-4.0) | 75.1 (-2.5) | 55.5 (-4.7) | 89.8 (-2.2) |
| w/ embedding | 59.5 (-6.3) | 75.8 (-3.0) | 44.0 (-6.9) | 74.0 (-3.6) | 52.1 (-8.1) | 88.0 (-4.0) |
| w/ LLM judge | 69.9 (+4.1) | 81.7 (+2.9) | 56.5 (+5.6) | 81.1 (+3.5) | 67.0 (+6.8) | 96.3 (+4.3) |

**Number of samples $k$.** As shown in Fig. 3a, across both uncertainty quantification and best-answer selection, RES outperforms alternative methods overall, and its gains are relatively insensitive to $k$ because it measures uncertainty via each answer's internal self-consistency rather than cross-answer consistency. At $k = 3$, RES already matches the performance of BSDETECTOR at $k = 8$, indicating stronger early performance and stability with fewer samples.

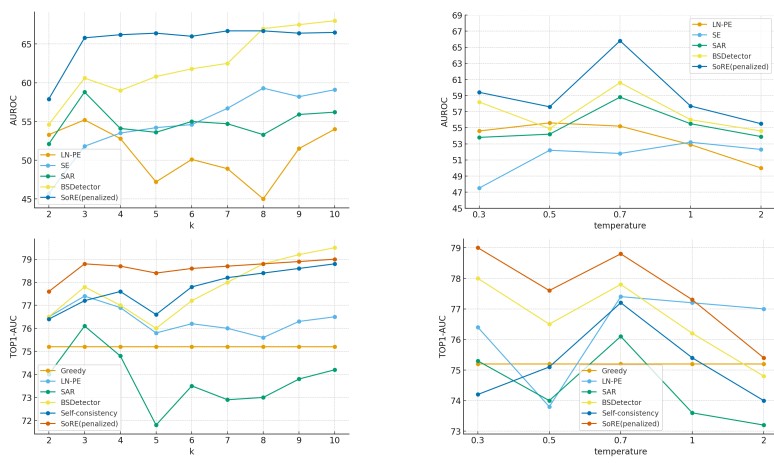

(a) Impact of the number of samples $k$.    (b) Impact of the sampling temperature.

Figure 3: Ablation study on the number of samples $k$ and temperature using GPT-4o-mini on StrategyQA, evaluated on AUROC and TOP1-AUC.

**Sampling temperature.**    As shown in Fig. 3b, RES achieves the best performance across all temperatures, and most methods peak around $t = 0.7$. At low temperatures, answers become more alike to one another and the internal reasoning paths also converge, reducing the separability between correct and incorrect outputs and thereby degrading the performance of both multi-sample and single-sample approaches. At high temperatures, increased diversity makes semantic clusters less stable; meanwhile, within an answer, explanations and reasoning paths tend to include irrelevant or incoherent content, which interferes with symmetry-based assessment and likewise harms performance.

**The role of structured prompting.**    Structured prompting divides the reasoning process into a three-part form; this is partly because the NLI model has limited input length and cannot ingest the full chain of reasoning, and partly because such structuring better aligns the reasoning and explanation paths. As show in Table 4, removing this structure **(w/o structured prompt)** results in incomplete sentences and weakened organization, which prevents the NLI model from making accurate judgments.

**What if we use alternatives to the NLI model?**    Replacing the NLI model with embedding cosine similarity **(w/ embedding)** likewise renders the uncertainty quantification less reliable, because cosine similarity captures only shallow semantic proximity. In contrast, NLI distinguishes subtle logical relations—entailment, neutrality, and contradiction—with directionality, thereby providing a more faithful representation of an answer's internal consistency. Using an LLM as the judge **(w/ LLM judge)** can further improve performance, as it aggregates multiple cues—including semantics, causality, and logical coherence—beyond sentence-pair relations. However, this introduces additional time and token costs, runs counter to our efficiency goals, and leads to unfair comparisons with other methods. Nevertheless, it remains a promising direction for future work.

## 5.4    TIME-COMPLEXITY ANALYSIS

We conduct experiments on the StrategyQA dataset using `Llama-3-8B` on a single NVIDIA GeForce RTX 4090 GPU. As shown in Fig. 4, we compare the average runtime per question and the detailed time composition across different methods to enable a finer-grained analysis of time complexity. Overall, RES achieves an average runtime between BSDETECTOR and LN-PE. The additional overhead compared to LN-PE baseline mainly comes from explanation-path generation (about 7s per call), while the cost of NLI-based judgment is negligible (about 0.2s). In contrast, BSDETECTOR although obtaining moderate performance gains through multiple self-reflection calls to the LLM, incurs substantially higher time overhead. In summary, RES strikes a favorable balance between effectiveness and efficiency.

(a) Total time per question ($k = 3$).

| Method | Time cost composition |
|---|---|
| RES | $k \times \text{gen} + k \times \text{explain} + 2k \times \text{NLI}$ |
| LN-PE | $k \times \text{gen}$ |
| SAR | $k \times \text{gen} + \binom{k}{2} \times \text{sim calculation}$ |
| BSDETECTOR ($r$ refl) | $k \times \text{gen} + (k \times r) \times \text{LLM reflection} + 2k \times \text{NLI}$ |

(b) Time cost composition of different methods.

Figure 4: Time complexity analysis of different methods.

| | *On May 9th, 2017 Jane bought 40 eggs. She ate one per day. Today she ran out of eggs. What is the date 24 hours later in MM/DD/YYYY?* | | | |
|---|---|---|---|---|
| **Cand.** | **Reasoning Path** | **Explanation Path** | **Conf** | **RES score** |
| 1 (Correct, **06/19/2017**) | 1) Premise/Evidence: Jane bought 40 eggs on May 9th, 2017, and ate one egg per day until she ran out. 2) Reasoning: If Jane bought 40 eggs and ate one per day, it would take her 40 days to finish all the eggs. Starting from May 9, 2017, adding 40 days takes us to June 18, 2017. Since she ran out of eggs on June 18, 2017, the date 24 hours later would be June 19, 2017. 3) Conclusion: \<final\>06/19/2017\</final\> | 1) Premise/Evidence: Jane bought 40 eggs on May 9th, 2017, and ate one egg per day. Since she consumed one egg each day, she would have 40 days of eggs. 2) Explanation: Starting from May 9th, 2017, if Jane eats one egg per day, she will run out of eggs after 40 days. Therefore, we can calculate the end date by adding 40 days to May 9th, resulting in June 17th, 2017, as her last day of egg consumption. When she runs out of eggs on June 17th, 2017, 24 hours later would be June 18th, 2017. 3) Conclusion: The final date after 24 hours on June 18th, 2017, is represented in the MM/DD/YYYY format as 06/19/2017. Thus, the answer is \<final\>06/19/2017\</final\>. | 0.9982 | **0.5953** |
| 2 (Incorrect, **06/18/2017**) | 1) Premise/Evidence: Jane bought 40 eggs on May 9th, 2017, and ate one egg per day until she ran out. 2) Reasoning: If Jane bought 40 eggs and ate one per day, it would take her 40 days to finish all the eggs. Starting from May 9th, 2017, adding 40 days brings us to June 17th, 2017. Running out of eggs on June 17th means the last egg was eaten on that day. The date 24 hours later is June 18th, 2017. 3) Conclusion: \<final\>06/18/2017\</final\> | 1) Premise/Evidence: Jane bought 40 eggs on May 9th, 2017, and ate one egg per day. Since she consumed one egg each day, she would have 40 days of eggs. 2) Explanation: Starting from May 9th, 2017, if Jane eats one egg per day, she will run out of eggs after 40 days. Counting 40 days from May 9th leads to June 18th, 2017. After running out of eggs on June 18th, 24 hours later would be June 19th, 2017. However, the question specifies the final date (when she ran out of eggs), which is June 18th. 3) Conclusion: The final date when Jane ran out of eggs is accurately represented as 06/18/2017. \<final\>06/18/2017\</final\> | 1.0000 | 0.4673 |

Table 5: Case study: Conf (token-logit confidence) favors incorrect candidate 2, while RES correctly identifies candidate 1 by evaluating reasoning–explanation symmetry.

## 5.5 CASE STUDY

**Why does Conf fail?** As shown in Table 5, in this case we extract the token-logits of the answer spans from the reasoning path. The model assigns high confidence to both candidate answers, including the incorrect ones, indicating an overconfidence issue (Kadavath et al., 2022). Such high confidence merely reflects certainty about the date format rather than logical correctness.

**Why does RES succeed?** In Candidate 1, the reasoning chain follows the logic "40 days after 05/09 is 06/18 → add one more day is 06/19," and the explanation chain consistently supports this reasoning. This full alignment yields a high symmetry score. In contrast, Candidate 2 shows a conflict: the reasoning chain assumes the last day is 06/17, while the explanation and conclusion claim 06/18, leading to logical inconsistency and a lower symmetry score. This demonstrates that RES is more robust than shallow confidence, and offering stronger interpretability for uncertainty quantification.

## 6 CONCLUSION AND FUTURE WORK

In this work, we proposed RES, a novel framework for uncertainty quantification in large language models that leverages reasoning–explanation symmetry. By evaluating the bidirectional entailment between structured reasoning and explanation paths, RES provides interpretable and reliable uncertainty estimates while significantly reducing dependence on multi-sample consistency. Extensive experiments across six datasets demonstrate that RES not only improves uncertainty quantification but also enhances best-answer selection under limited sampling budgets.

Future work will focus on two main directions: (i) enhancing the performance of RES on commonsense reasoning tasks; (ii) further reducing time and token costs, such as employing lightweight explanation generation, with the ultimate goal of advancing more reliable and efficient LLM systems.

### Ethics Statement

Use unnumbered third level headings for the acknowledgments. All acknowledgments, including those to funding agencies, go at the end of the paper. This study aims to enhance the reliability and safety of LLM outputs, with a core goal that holds significant societal value. The "Reasoning-Explanation Symmetry" (RES) method we propose is designed to assist in hallucination detection and selective generation by quantifying uncertainty, thus reducing the risk of generating incorrect or harmful information. We must emphasize that a key limitation of the RES method is that it measures logical "consistency" rather than "factual correctness" or "social fairness." An answer containing stereotypes or biases may still be rated highly (i.e., with low uncertainty) by our method if its reasoning and explanation paths are logically symmetric. Therefore, RES should not be viewed as a tool for eliminating bias. Its evaluation results should be used in conjunction with other tools, such as fact-checking and bias detection, to ensure the fairness and accuracy of the final output.

### Reproducibility Statement

We are committed to ensuring the full reproducibility of this research work. To this end, we provide detailed descriptions of the algorithms, experimental setups, and implementation details in the paper.

- **Core Algorithm**: A comprehensive description of the proposed "Reasoning-Explanation Symmetry" (RES) framework, including the three stages of structured sampling, paired explanation generation, and symmetry scoring, is provided in the methodology section (Section 3) and Appendix A.

- **Experimental Setup**: The complete experimental setup is detailed in Section 4.

- **Datasets**: All datasets used in this work are listed in Section 4.1 and Table 1, with further usage details provided in Appendix B.

- **Evaluation Metrics**: The evaluation metrics for uncertainty quantification (AUROC) and best-answer selection (TOP1-AUC) are clearly defined in Section 4.2.

- **Hyperparameters and Models**: All hyperparameters used in the experiments (e.g., sampling temperature $t = 0.7$, sample count $k = 3$), as well as the specific versions of the large language models and NLI models employed, are described in Section 4.4.

- **Computing Environment**: The computational hardware environment used for the time complexity analysis is mentioned in Section 5.4.

- **Code and Resources**: To facilitate further reproducibility, the full source code will be provided in the supplementary materials.

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

## A  PROMPT TEMPLATES

### A.1  REASONING PROMPT

```
You are a careful reasoner. Think first, then answer.
Rules:
1) Keep the content concise; no placeholders such as: No valid
   answer, N/A, Unknown, None.
2) The LAST line must contain ONLY the final answer wrapped with
   <final>YOUR ANSWER</final>.

Output format: Use EXACTLY these sections and headers:
1) Premise/Evidence:
2) Reasoning:
3) Conclusion:

Question: {question}
Answer:
```

## A.2 EXPLANATION PROMPT

```
You already answered the question. Your ONLY task is to explain why
   the given answer is correct.
Rules:
1) DO NOT change, contradict, paraphrase, or propose any other
   answer.
2) Keep the content concise; no placeholders such as: No valid
   answer, N/A, Unknown, None.
3) The LAST line must contain ONLY the ORIGINAL given answer
   wrapped with <final>YOUR ANSWER</final>.

Output format: Use EXACTLY these sections and headers:
1) Premise/Evidence:
2) Explanation:
3) Conclusion:

Question: {question}
Given Answer: {answer_text}
Explanation:
```

## B DATA DETAILS

Due to resource limitations, we selected approximately 3,000 QA pairs from the CoQA dev split and about 3,000 QA pairs from the TriviaQA train split. For TriviaQA, we did not use the accompanying reference documents, as we were more interested in examining the capability of RES to distinguish commonsense reasoning without relying on external knowledge. The Date Understanding task was originally in a multiple-choice format, but we removed the options and required the LLM to independently derive the answer, since our goal was to evaluate RES on more challenging numerical reasoning tasks. Many questions in CoQA are incomplete and depend on the conversational history for correct answers; therefore, we included the previous QA history in the prompt and used the first generated answer for each question.

Below we provide the few-shot prompts used for each task:

**TriviaQA Few-shot Prompt**

```
Examples:

Question: In America, what became the 49th state to enter the union
    in 1959?
Answer:
1) Premise/Evidence:
On January 3, 1959, following the signing of the Alaska Statehood
    Act by President Dwight D. Eisenhower, Alaska was officially
    admitted to the United States.
2) Reasoning:
The question requires identifying the state that fulfilled two
    conditions: being the 49th to join the union and doing so in
    the year 1959.
3) Conclusion:
<final>Alaska</final>
```

**CoQA Few-shot Prompt**

```
Examples:

Story: "Trinity College is a constituent college of the University
    of Cambridge in England..."

Question: What kind of school is this?
Answer:
1) Premise/Evidence:
The text says "Trinity College is a constituent college of the
    University of Cambridge in England."
2) Reasoning:
The term "constituent college" indicates it is part of a larger
    university.
3) Conclusion:
<final>a constituent college</final>
```

**Date Understanding Few-shot Prompt**

```
Examples:

Question: Today is Christmas Eve of 1937. What is the date tomorrow
    in MM/DD/YYYY?
Answer:
1) Premise/Evidence:
Today's date is December 24, 1937.
2) Reasoning:
Add one day -> 12/25/1937.
3) Conclusion:
<final>12/25/1937</final>
```

**Multistep Arithmetic Few-shot Prompt**

```
Examples:

Question: ((-1 + 2 + 9 * 5) - (-2 + -4 + -4 * -7)) =
Answer:
1) Premise/Evidence:
We are asked to compute the expression.
2) Reasoning:
- First part: -1 + 2 + 9 * 5 = 46
- Second part: -2 + -4 + -4 * -7 = 22
- Subtract: 46 - 22 = 24
3) Conclusion:
<final>24</final>
```

**MultiRC Few-shot Prompt**

```
Examples:

Passage: Carl the robot was missing a tire and a sun gatherer...

Question: What did Carl need before going to the lab?
Candidate option: Tire
Answer:
1) Premise/Evidence:
The passage states Carl was missing a tire.
2) Reasoning:
"Tire" was explicitly mentioned as needed.
3) Conclusion:
<final>Yes</final>
```

**StrategyQA Few-shot Prompt**

```
Examples:

Question: Is it unusual to play Happy hardcore music at a funeral?
Answer:
1) Premise/Evidence:
Happy hardcore is a fast-paced, upbeat electronic music genre.
2) Reasoning:
It contrasts with the solemn mood of funerals.
3) Conclusion:
<final>Yes</final>
```

## C    SENSITIVITY TO CORRECTNESS MEASURES

Table 6: Sensitivity of RES(mean) to ROUGE-L threshold on TriviaQA and CoQA (GPT-4o-mini).

| Dataset | Metric | 0.3 | 0.5 | 0.7 |
|---------|--------|-----|-----|-----|
| TriviaQA | AUROC | 84.2 | 80.0 | 76.1 |
| | TOP1-AUC | 68.2 | 64.5 | 62.0 |
| CoQA | AUROC | 82.2 | 78.0 | 73.2 |
| | TOP1-AUC | 56.4 | 50.8 | 49.0 |

The threshold used to determine whether an answer is correct can substantially affect evaluation. As shown in Table 6, increasing the ROUGE-L threshold leads to decreases in both AUROC and TOP1-AUC. The main reason is that, in free-form QA, answers exhibit greater variability and longer sequences, whereas our prompt asks the model to produce a concise final answer. When the correctness threshold is set too high, the extracted answer may have limited overlap with the gold label, resulting in degraded detection performance.

## D    SENSITIVITY TO PENALTY COEFFICIENT $\lambda$

Table 7: Ablation on the penalty coefficient $\lambda$ for RES(PENALIZED) (GPT-4o-mini).

| Dataset | Metric | 0.5 | 0.8 | 1.0 | 1.2 | 1.5 |
|---|---|---|---|---|---|---|
| MultiRC | AUROC | 58.0 | 58.3 | 58.1 | **58.2** | 58.0 |
| | TOP1-AUC | 85.5 | 85.7 | 85.6 | **85.8** | 85.5 |
| Date Understanding | AUROC | 61.0 | 61.2 | 61.1 | **61.3** | 61.0 |
| | TOP1-AUC | 73.7 | 73.9 | 73.8 | **74.0** | 73.6 |
| Multistep Arithmetic | AUROC | 59.9 | 60.1 | 60.0 | **60.2** | 59.9 |
| | TOP1-AUC | 91.8 | 91.9 | 91.8 | **92.0** | 91.7 |
| StrategyQA | AUROC | 65.7 | 65.8 | 65.6 | **65.8** | 65.5 |
| | TOP1-AUC | 78.6 | 78.7 | 78.5 | **78.8** | 78.4 |
| TriviaQA | AUROC | 79.8 | 79.9 | 79.7 | 80.0 | **80.4** |
| | TOP1-AUC | 64.4 | 64.5 | 64.3 | 64.6 | **65.2** |
| CoQA | AUROC | 77.4 | 77.5 | 77.3 | **77.6** | 77.2 |
| | TOP1-AUC | 50.7 | 50.9 | 50.6 | **51.0** | 50.6 |

As shown in Table 7, most datasets achieve their best performance around $\lambda = 1.2$, indicating that the contradiction probability has strong discriminative power in uncertainty quantification and thus should be assigned a relatively large coefficient. If $\lambda$ is too small, contradictions are under-penalized, causing ambiguous or partially incorrect answers to receive inflated scores and degrading ranking and selection performance.

## E    THE USE OF LARGE LANGUAGE MODELS

This manuscript used a large language model only for light editorial support—namely grammar and spelling checks, minor language polishing, and table formatting. The LLM did not generate scientific content, results, analyses, or claims. All edits were reviewed by the authors, and the authors remain fully responsible for the final text.

