# OpenReview forum: "Uncertainty Quantification via Reasoning–Explanation Symmetry in LLMs"
_ICLR.cc/2026/Conference — ICLR 2026 Conference Withdrawn Submission_

### Official Review · Reviewer_EXuc · 2025-10-16

**Soundness:** 2
**Presentation:** 3
**Contribution:** 2
**Rating:** 4
**Confidence:** 4

**Summary:**

This paper proposes to use the Reasoning–Explanation Symmetry scores to quantify the uncertainty of an LLM from its answers and reasoning chains. Specifically, the authors sampled three answers with reasoning chains from the language model to a question, and then prompt the language model to explain the answers given the questions. The authors use the RoBERTa model pre-trained on the NLI task to calculate the entailment probabilities of the reasoning–explanation pairs and aggregate them as an evaluation of the confidence of LLM to its answer.

**Strengths:**

The paper is straightforward and easy-to-follow. The proposed idea of using Reasoning–Explanation Symmetry scores to evaluate the uncertainty/confidence of a language model is interesting and could be inspiring to some of the researchers in the community. Overall, the proposed method, experiment setup and result analysis make sense. The proposed idea does introduce certain performance gain on some datasets.

**Weaknesses:**

The motivation of this paper is not well-defined. The authors claims "uncertainty serves as a signal for hallucination detection and selective generation" but I personally do not understand how AUROC and TOP1-AUC are good metrics for hallucination detection and selective generation. I'd appreciate some explanation here.

The authors also claim the proposed method can "quantify uncertainty without multiple sampling". But from the following description it looks like the method sampled 2*k (=6) responses almost independently from the language model after all. I don't understand why this is not "multiple sampling".

In lines 034 the authors claim "studies have shown that these measures (token probs) correlate weakly with generation quality". This is not the case. Also I did not find where the reference (Chen et al., 2023) made or supported this claim. The authors may checkout some recent works [1--3] discussing related methods.

Regarding the methodology, I think it is more like a minor tweak to directly generative 2k=6 answers and calculate the pair-wise entailment probabilities. The authors can add related ablation studies to support the effectiveness of the "explanation" step.

Regarding the selection of datasets, I wonder why the authors choose a subset instead of using the entire BBH dataset. Also some benchmarks from other domains such as MATH or GSM8k can be adopted in the experiments.

Figure 4b indicates that the latency of RES is at least 2X of LN-PE but Figure 4a shows the opposite. Considering that the case in Table 5 shows longer explanation paths I would appreciate some explain on this difference.

Table 2 shows that the AUROC of some methods falls below 0.5 which indicates that incorrect answers are associated with higher confidence. This deviates from the results in their original papers and recent survey papers [3]. Please explain.

Misssing reference in line 260.

[1] Zhang, Tunyu, et al. "Token-Level Uncertainty Estimation for Large Language Model Reasoning." arXiv preprint arXiv:2505.11737 (2025).
[2] Li, Yinghao, et al. "Language model uncertainty quantification with attention chain." arXiv preprint arXiv:2503.19168 (2025).
[3] Abbasli, Toghrul, et al. "Comparing Uncertainty Measurement and Mitigation Methods for Large Language Models: A Systematic Review." arXiv preprint arXiv:2504.18346 (2025).

**Questions:**

Please refer to "Weaknesses"

---

> ### Author Response · Authors · 2025-11-13
>
> We sincerely thank the reviewer for the thorough and constructive feedback. Below we provide point-by-point responses.
> > **R1 | The motivation of this paper is not well-defined. The authors claims "uncertainty serves as a signal for hallucination detection and selective generation" but I personally do not understand how AUROC and TOP1-AUC are good metrics for hallucination detection and selective generation.**
>
> We apologize for the confusion. When we say that *“uncertainty can serve as a signal for hallucination detection and selective generation”*, we refer to the **scalar uncertainty score** produced by our method for each model output, not to AUROC or TOP1-AUC themselves. In our framework, the uncertainty score \(u(x)\) is the quantity that a deployed system would actually use—for example, by thresholding \(u(x)\) to flag hallucinations or to abstain from answering when uncertainty is high.
>
> AUROC and TOP1-AUC are **purely evaluation metrics**, used only offline to measure how informative this uncertainty score is. For hallucination detection, AUROC summarizes how well \(u(x)\) separates hallucinated from correct answers across all possible thresholds, i.e., the probability that a randomly drawn hallucinated answer receives a higher uncertainty score than a randomly drawn correct one. For selective generation, TOP1-AUC (risk/accuracy–coverage area) summarizes the trade-off induced by thresholding \(u(x)\) at different levels: a better uncertainty signal leads to higher accuracy (or lower risk) for the same coverage.
>
> We will revise the introduction to explicitly distinguish between (i) the **uncertainty score as the operational signal** used for detection/abstention, and (ii) **AUROC / TOP1-AUC as standard evaluation measures** that quantify the quality of this signal, so as to avoid the impression that the metrics themselves are used as decision signals.
>
> ---
>
> > **R2 | The authors also claim the proposed method can "quantify uncertainty without multiple sampling". But from the following description it looks like the method sampled 2*k (=6) responses almost independently from the language model after all.**
>
> There are two reasons why the evaluation was not performed under the single-sample condition k=1:
> 1. For the UQ task, strong baseline methods (INSIDE, BSDector) all require the number of samples to be greater than 1, so we did not evaluate the single-sample case for a complete comparison
> 2. For the BEST-ANSWER SELECTION task, the goal is to select the most accurate answer from multiple candidates, which is also not suitable for single-sample evaluation.
>
> We will add the single-sample evaluation results to the ablation study in the revised version:
>
> | k | LN-PE | SE | SAR | BSDetector | **RES(penalized)** |
> | :---: | :---: | :---: | :---: | :---: | :---: |
> | **1** | **41.5** | N/A | **38.7** | N/A | **61.8** |
> | 2 | 53.3 | 45.7 | 52.1 | 54.6 | 57.9 |
> | 3 | 55.2 | 51.8 | 58.8 | 60.6 | 65.8 |
> | 4 | 52.8 | 53.5 | 54.1 | 59.0 | 66.2 |
> | 5 | 47.2 | 54.2 | 53.6 | 60.8 | 66.4 |
> | 6 | 50.1 | 54.6 | 55.0 | 61.8 | 66.0 |
> | 7 | 48.9 | 56.7 | 54.7 | 62.5 | 66.7 |
> | 8 | 45.0 | 59.3 | 53.3 | 67.0 | 66.7 |
> | 9 | 51.5 | 58.2 | 55.9 | 67.5 | 66.4 |
> | 10 | 54.0 | 59.1 | 56.2 | 68.0 | 66.5 |
>
> It can be seen that LN-PE and SAR severely degrade with a single sample, while RES's performance remains stable, showing its significant advantage in the single-sample scenario.
>
> ---
>
> > **R3 | In lines 034 the authors claim "studies have shown that these measures (token probs) correlate weakly with generation quality". This is not the case.**
>
> Thank you for the suggestion. Our statement here was indeed inaccurate. We meant to convey that token probability-based methods are relatively and clearly inferior to semantic methods (Semantic Density (Qiu & Mikkulainen 2024)). We will remove the inappropriate citation and revise this statement.

---

> ### Author Response · Authors · 2025-11-13
>
> > **R4 | Regarding the methodology, I think it is more like a minor tweak to directly generative 2k=6 answers and calculate the pair-wise entailment probabilities. The authors can add related ablation studies to support the effectiveness of the "explanation" step.**
>
> First, as mentioned in R2, k=3 is only for the sake of experimental completeness, and our method is still applicable when k=1.
>
> Second, the explanation chain is a different semantic trajectory induced by the "restate-justify" instruction, ensuring it forms a true dual perspective with the reasoning chain, amplifying the conflict signal resulting from reasoning failure. Generating 2k reasoning chains cannot utilize the R/E symmetry, cannot detect reverse entailment, and performs poorly with a small number of samples.
>
> Finally, according to your suggestion, we will supplement the following ablation study, comparing the results of letting the model generate 2k answers without generating explanations. It can be seen that after removing the explanation step, RES's performance significantly drops because the evaluation method at this point is closer to traditional SE.
>
> | Method | StrategyQA AUROC | StrategyQA TOP1-AUC | CoQA AUROC | CoQA TOP1-AUC | Multistep Arithmetic AUROC | Multistep Arithmetic TOP1-AUC |
> | :--- | ---: | ---: | ---: | ---: | ---: | ---: |
> | **RES (penalized)** | **65.8** | **78.8** | **77.6** | **51.0** | **60.2** | **92.0** |
> | w/o structured prompt | 63.0 (−2.8) | 77.3 (−1.5) | 75.1 (−2.5) | 46.9 (−4.1) | 55.5 (−4.7) | 89.8 (−2.2) |
> | w/ embedding | 59.5 (−6.3) | 75.8 (−3.0) | 74.0 (−3.6) | 44.0 (−7.0) | 52.1 (−8.1) | 88.0 (−4.0) |
> | w/ LLM judge | 69.9 (+4.1) | 81.7 (+2.9) | 81.1 (+3.5) | 56.5 (+5.5) | 67.0 (+6.8) | 96.3 (+4.3) |
> | **w/o explanation** | 46.5 (−19.3) | 73.5 (−5.3) | 67.8 (−9.8) | 45.0 (−6.0) | 49.5 (−10.7) | 88.8 (−3.2) |
>
> ---
>
> > **R5 | Regarding the selection of datasets, I wonder why the authors choose a subset instead of using the entire BBH dataset. Also some benchmarks from other domains such as MATH or GSM8k can be adopted in the experiments.**
>
> Because BBH contains 23 tasks, experimenting on the entire dataset would be too time-consuming, and some of the tasks are open-ended or subjective (e.g., Movie Recommendation, Ruin Names) and are not suitable for uncertainty evaluation. We used Multi-Step Arithmetic, which, like MATH or GSM8k, is used to evaluate the model's arithmetic capability, but the latter's difficulty is lower. We aimed to test the model's performance on more complex tasks.
>
> ---
>
> > **R6 | Figure 4b indicates that the latency of RES is at least 2X of LN-PE but Figure 4a shows the opposite. Considering that the case in Table 5 shows longer explanation paths I would appreciate some explain on this difference.**
>
> In Figure 4a, the time taken by RES is 49.1s, and for LN-PE it is 25.7s, which is consistent with the analysis in Figure 4b.
>
> ---
>
> > **R7 | Table 2 shows that the AUROC of some methods falls below 0.5 which indicates that incorrect answers are associated with higher confidence. This deviates from the results in their original papers and recent survey papers [3]. Please explain.**
>
> Firstly, it should be clarified that the AUROC values below 0.5 are the results of the two early baselines, LN-PE and SE, on the MultiRC, Date Understanding, and Multistep Arithmetic datasets. Neither the original papers nor the survey by Toghrul Abbasli et al. used these specific datasets, so the claim of "deviation in results" does not hold.
>
> As for why AUROC values below 0.5 appear on these datasets, we believe it is because these datasets require complex reasoning and are significantly more difficult than simple reading comprehension and QA tasks like TriviaQA and CoQA. However, the model did not lower its confidence despite the increased difficulty.
>
> For the commonly used TriviaQA and CoQA datasets, our reported results are generally consistent with those in the original paper (**SEMANTIC UNCERTAINTY: LINGUISTIC INVARIANCES FOR UNCERTAINTY ESTIMATION IN NATURAL LANGUAGE GENERATION, Lorenz Kuhn et al. — Figure 2**). Any potential deviation is mainly due to the difference in the number of samples used (k=10 in the original paper).
>
> ---
>
> > **R8 | Missing reference in line 260.**
>
> We will supplement the missing citation in the corresponding place.

---

> > ### Comment · Reviewer_EXuc · 2025-11-14
> > **Claims should be supported**
> >
> > Q1 -> If your evaluation metrics do not support your claim in Intro, do not make the claim.
> >
> > Q2 -> This is another significant and fatal discrepancy between your claim and your experiment evidences. The k=1 results should be your main results rather than an "ablation study". Missing it from the original draft basically sentenced its death.
> >
> > Q3 -> I don't understand how you could randomly find a paper to "support" the ungrounded claim. You should treat your paper more seriously.
> >
> > Q5 -> No this is not a good answer. If you cannot finish the experiments in time you can always submit the draft to another venue. The entire BBH, MATH, and GSM8k are standard benchmarks and more widely used in other articles. The excuses for excluding them are not reasonable.
> >
> > Q6 -> If I can still do math I think 25.7 * 2 = 51.4 > 49.1? Correct me if I'm wrong.
> >
> > Q7 -> That's why I suggested to run experiments on the entire dataset and other widely applied datasets.
> >
> > I appreciate your answer to Q4.
> >
> > Overall, the quality of the draft is concerning due to numerous unsupported claims. I'll keep the my current score, 4, for personal reasons but this draft is a clear rejection to me.

---

### Official Review · Reviewer_8To9 · 2025-10-21

**Soundness:** 2
**Presentation:** 3
**Contribution:** 2
**Rating:** 2
**Confidence:** 4

**Summary:**

The paper introduces Reasoning-Explanation Symmetry (RES) to quantify the consistency of LLM answers for hallucination detection.
For a given question, first a structured reasoning is generated, followed by a final answer.
Then given the question and the answer, a structured explanation is generated.
An NLI model is this used to compare reasoning and explanation, computing a consistency score based on the predictions of the NLI model.
The resulting score is utilized for hallucination detection and best-answer selection tasks.
RES is efficient at relatively small sampling counts and the extra generated reasoning and explanation offers interpretability in the generated answer.

**Strengths:**

- The approach of structured reasoning and explanations to cope with the limitations of the NLI model is very practical also beyond this work.
- A wide variety of datasets were used for the evaluation.
- Having reasoning and explanation as additional tool for interpretability warrants the additional computational costs incured by using this method for hallucination detection.
- The treatment of prior work (except the very closely related BSDetector which should be discussed in more detail) is extensive and the selection of baselines is reasonable.
- Interesting ablation studies, especially the role of structured prompting and NLI model alternatives.

**Weaknesses:**

- Unfortunately, I have to disagree on a fundamental premise of the paper, namely that consistency or the lack thereof is the same as uncertainty in the prediction. Let me give a counterexample to this intuition. For a given input the model could be uncertain about the reasoning as there are two possible ways to answer the question, but the answer itself is certain given one of the two reasoning paths. Thus for a given answer, the model is also very certain about the explanation, indicating high certainty overall. However, would one generate a second time, chances could be 50:50 that the other reasoning would be applied, leading to a very different answer, yet again a consistent explanation would be generated for this answer. In essence, the suggested relationship may only hold if there is only a single true answer to the question. However, I still find the premise of the approach convincing to generally assess the reliability of LLM answers and detecting hallucinations, yet I would prefer if it would be marketed as such. Alternatively, a more convincing argument why consistency and uncertainty should be the same or conditions under which this may hold would be necessary.
- The idea of the paper is very close to the BSDetector algorithm in using the NLI model for the final score. This would warrant a more extensive treatment and delineation from RES in the related work section.
- One of the main selling points in the abstract is that RES does not do multiple sampling (line 16/17), which is not how it is evaluated in the experiments where k=3 samples are drawn, later on there is also an ablation with a minimum of k=2. The method is never evaluated on a single sample k=1. Note that also baseline methods such as LN-PE or SAR (TokenSAR) could be evaluated on just a single sample. I would tone down this argument or remove it completely. What the results in the ablation study in section 5.3 show is, that is more effectively utilizes samples (k), yet one has to bear in mind that samples for RES are more expensive than for the baseline methods.
- The actual score calculation is handed off to a secondary BERT model trained on the NLI task, thus is completely reliant on the fact that the NLI model generalizes to the particular domain that RES should be used in. This limitation should at least be acknowledged to a sufficient extent, as done e.g. in the BSDetector paper.
- The weakness stated in line 45-49, namely that semantic metrics are only useful on a question rather than a answer is not entirely true. The work of Qiu and Mikkulainen (2024) tackles excactly that issue.
- Using ROUGE-L as a correctness metric has recently been shown to be problematic by various independent papers on the matter (Santilli et al., Ielanskyi et al. and Janiak et al.). Their findings suggest that using LLM-as-a-judge approaches, best with different prompting strategies and aggregation, would lead to much more dependable results.
- Furthermore, PRR (Malinin & Gales) would be preferable as an evaluation metric, complimenting the AUROC used so for in the results. PRR is usually preferable, as it is less impacted by the correctness of predictions by the LLM.

Remarks:
- While reading, I found it unintuitive that the resulting score is higher for more reliable (certain) samples, which is opposite of what is standard for a general uncertainty score. For example line 254 reads strange due to that, where samples with highest UQ score are selected for best-choice accuracy. For hasty readers, this might be irritating.
- Is the SoRE method in Figure 3 an earlier naming of RES? I couldn't find SoRE referenced anywhere else and RES results would be missing otherwise.
- In line 39, the SDLG method is called "Semantically Diverse *Language* Generation"

---
Qiu, Mikkulainen (2024) Semantic Density: Uncertainty Quantification for Large Language Models through Confidence Measurement in Semantic Space, NeurIPS

Santilli et al. (2025) Revisiting Uncertainty Quantification Evaluation in Language Models: Spurious Interactions with Response Length Bias Results, ACL

Ielanskyi et al. (2025) Addressing Pitfalls in the Evaluation of Uncertainty Estimation Methods for Natural Language Generation, ArXiv

Janiak et al. (2025) The Illusion of Progress: Re-evaluating Hallucination Detection in LLMs, EMNLP

Malinin, Gales (2021) Uncertainty Estimation in Autoregressive Structured Prediction, ICLR

**Questions:**

- I would be curious about using - mean(c_{i,fwd}, c_{i,bwd}) as scoring function, would this be beneficial over the *penalized* score?
- Regarding the case study, how can the confidence scores in Table 5 be so high? This would imply that every token of the answer is predicted with (near) certainty - a probability very close to 1 - or how is the confidence calculated for this example?
- Interestingly, the performance of RES is weakest on the TriviaQA and CoQA datasets, which are pretty standard in prior work on uncertainty estimation for LLMs. Do you have any insights into why the method struggled on those datasets or is it rather that baselines were already tuned well for them?
- Recently, Aichberger et al. (2024) showed that the likelihood of the greedily decoded sequence can be a very strong measure of uncertainty. Due to it being basically for free, I would like to know how well it performs as uncertainty measure on the considered tasks.
- In Figure 3, what temperature was used in (a) and what k in (b)? I would guess t=0.7 and k=3 from the description at the beginning of the experimental section, but it should be stated when describing this experiment.

---
Aichberger, Schweighofer, Hochreiter (2024) Rethinking Uncertainty Estimation in Natural Language Generation, ArXiv

---

> ### Author Response · Authors · 2025-11-12
> **Explanation of the basic premise**
>
> Thank you for your suggestion. Your perspective introduces a new viewpoint to our paper, particularly the counterexample in the first point, which we believe precisely explains why our method is superior to methods based on cross-sample consistency.
>
> ---
>
> > **R1 | Regarding the premise of "Consistency = Uncertainty"**
>
> We agree with your point: in a scenario where a question has multiple correct answers, the **question-level uncertainty** may indeed be high, but this is not equivalent to the model's uncertainty about the reliability of a single answer. Our method focuses on the **answer-level uncertainty**, rather than the uncertainty of the question itself. The difference between the two is:
>
> * Higher question-level uncertainty implies greater **answer diversity** for a fixed question.
> * Higher answer-level uncertainty implies greater **diversity in the reasoning process** for a fixed answer.
>
> This exactly corresponds to the counterexample you raised: question-level uncertainty reflects answer diversity, but diversity does not equate to accuracy, because diversity is also high when a question has multiple correct answers. Therefore, RES avoids this misjudgment by using an answer-level metric, specifically, **"whether the model is self-consistent when generating this particular answer."**
>
> We will clarify the distinction between question-level and answer-level uncertainty in the paper and highlight it as a major selling point for RES.
>
> ---
>
> > **R3 | A main claim in the abstract is that RES does not perform multi-sampling...**
>
> > **R5 | Semantic metrics are only useful on a question rather than a answer is not entirely true (Qiu & Mikkulainen, 2024).**
>
> Building upon R1, we would like to address these two points first.
>
> Regarding R5, we acknowledge the omission of relevant literature (Qiu, Mikkulainen (2024)) and will revise the description in lines 45-49. However, it must be noted that the study by Qiu and Mikkulainen still relies on reference answers obtained from multiple sampling. Although the resulting uncertainty score is answer-level, it still reflects **cross-sample consistency** rather than **internal consistency of the answer**. Thus, this work is fundamentally different from the motivation we discussed in R1.
>
> Regarding R3, there are two reasons why we did not initially evaluate the single-sample case ($k=1$):
>
> 1. For the **UQ task**, strong baseline methods (INSIDE, BSDETECTOR) all require the number of samples to be greater than one ($k>1$), so we omitted the single-sample evaluation to ensure a complete comparison.
> 2. For the **BEST-ANSWER SELECTION task**, the goal is to select the most accurate answer from multiple candidates, which is also not suitable for single-sample evaluation.
>
> We will add the following single-sample evaluation results to the ablation study in the revised version:
>
> | k | LN-PE | SE | SAR | BSDetector | **RES(penalized)** |
> | :---: | :---: | :---: | :---: | :---: | :---: |
> | **1** | **41.5** | N/A | **38.7** | N/A | **61.8** |
> | 2 | 53.3 | 45.7 | 52.1 | 54.6 | 57.9 |
> | 3 | 55.2 | 51.8 | 58.8 | 60.6 | 65.8 |
> | 4 | 52.8 | 53.5 | 54.1 | 59.0 | 66.2 |
> | 5 | 47.2 | 54.2 | 53.6 | 60.8 | 66.4 |
> | 6 | 50.1 | 54.6 | 55.0 | 61.8 | 66.0 |
> | 7 | 48.9 | 56.7 | 54.7 | 62.5 | 66.7 |
> | 8 | 45.0 | 59.3 | 53.3 | 67.0 | 66.7 |
> | 9 | 51.5 | 58.2 | 55.9 | 67.5 | 66.4 |
> | 10 | 54.0 | 59.1 | 56.2 | 68.0 | 66.5 |
>
> It can be observed that LN-PE and SAR severely **degrade** in the single-sample case; however, RES's performance remains stable, demonstrating its significant advantage in the single-sample scenario. We believe this advantage is sufficient to justify the minimal additional sample cost compared to the baselines.
>
> In summary, although previous methods could either use single-sample evaluation or perform answer-level measurement, **only RES combines both**, achieving answer-level single-sample evaluation.

---

> ### Author Response · Authors · 2025-11-12
>
> >**R4 | The actual score calculation is handed off to a secondary BERT model trained on the NLI task...**
>
> Thank you for your suggestion. We have also noted the limitations associated with relying on NLI for scoring. In Table 4, we provide a comparison using (i) embedding similarity and (ii) an LLM-judge as alternatives to NLI. We suggest lightweight calibration or distillation for the target domain. We will add a new section to discuss the limitations of RES.
>
> ---
>
> >**R6 |  Using LLM-as-a-judge approaches**
>
> Thank you for your suggestion. We have supplemented the results using Llama3-8B to generate replies and GPT-5 as the evaluation model. Other parameters still adhere to the paper's original settings. It can be observed that after adopting an LLM as the judgment criterion, RES's performance slightly decreases (the difference from the ROUGE-L result is in parentheses) but remains generally stable, and even surpasses BSDetector on CoQA. We will provide the complete experimental results in the revised version.
>
> | Dataset | LN-PE | SE | SAR | BSDetector | INSIDE | RES(min) | RES(mean) | RES(penalized) |
> | :--- | ---: | ---: | ---: | ---: | ---: | ---: | ---: | ---: |
> | MultiRC | 35.7 (−8.4) | 41.3 (−7.6) | 45.3 (−6.9) | 51.2 (−2.0) | 44.5 (−3.5) | 51.0 (−1.2) | **57.9 (−1.0)** | 57.1 (−0.9)|
> | Date Understanding | 33.1 (−7.9) | 32.8 (−8.1) | 38.8 (−7.2) | 51.4 (−2.5) | 40.4 (−3.0) | 57.5 (−1.5) | 58.0 (−1.7) | **58.3 (−1.4)** |
> | StrategyQA | 43.8 (−6.4) | 41.0 (−6.8) | 45.7 (−6.0) | 51.4 (−1.7) | 49.9 (−2.2) | 51.3 (−1.1) | **53.1 (−1.0)** | 52.7 (−1.0) |
> | TriviaQA | 55.3 (−9.2) | 57.9 (−8.8) | 65.4 (−8.1) | **77.2 (−2.3)** | 67.2 (−3.4) | 69.9 (−1.6) | 73.4 (−1.9) | 74.8 (−1.7) |
> | CoQA | 68.3 (−7.1) | 66.1 (−6.5) | 70.0 (−6.9) | 76.5 (−1.7) | 74.8 (−2.0) | 75.1 (−0.9) | **76.7 (−1.1)** | 75.5 (−1.0) |
>
> ---
>
> >**R7 | PRR (Malinin & Gales) would be preferable as an evaluation metric**
>
> In the paper, we use another metric, **TOP1-AUC**, instead of PRR. Both metrics can be used to evaluate the relationship between the uncertainty score and the actual output quality, but TOP1-AUC is more intuitive. In Table 3, we provide examples of selecting answers based on the uncertainty score. By comparing the results of RES with greedy decoding, we can see that RES helps improve the accuracy of the response.
>
> | Dataset | Model | Greedy | RES(min) | RES(mean) | RES(penalized) |
> | :--- | :--- | ---: | ---: | ---: | ---: |
> | MultiRC | GPT-4o-mini | 84.2 | 85.0 (+0.8) | 85.6 (+1.4) | 85.8 (+1.6) |
> | | Qwen3-8B | 84.6 | 85.8 (+1.2) | 86.9 (+2.3) | 87.2 (+2.6) |
> | | Llama3-8B | 72.4 | 76.0 (+3.6) | 76.2 (+3.8) | 74.0 (+1.6) |
> | Date Understanding | GPT-4o-mini | 70.4 | 71.5 (+1.1) | 73.2 (+2.8) | 74.0 (+3.6) |
> | | Qwen3-8B | 76.4 | 79.6 (+3.2) | 81.8 (+5.4) | 82.0 (+5.6) |
> | | Llama3-8B | 42.8 | 47.6 (+4.8) | 49.2 (+6.4) | 49.6 (+6.8) |
> | Multistep Arithmetic | GPT-4o-mini | 88.4 | 90.4 (+2.0) | 92.4 (+4.0) | 92.0 (+3.6) |
> | | Qwen3-8B | 98.8 | 99.2 (+0.4) | 99.6 (+0.8) | 99.6 (+0.8) |
> | | Llama3-8B | 39.6 | 44.0 (+4.4) | 44.8 (+5.2) | 44.8 (+5.2) |
> | StrategyQA | GPT-4o-mini | 76.2 | 77.5 (+1.3) | 78.9 (+2.7) | 78.8 (+2.6) |
> | | Qwen3-8B | 76.8 | 78.0 (+1.2) | 78.6 (+1.8) | 78.6 (+1.8) |
> | | Llama3-8B | 64.2 | 64.6 (+0.4) | 66.0 (+1.8) | 67.6 (+3.4) |
> | TriviaQA | GPT-4o-mini | 63.6 | 64.0 (+0.4) | 64.5 (+0.9) | 64.6 (+1.0) |
> | | Qwen3-8B | 44.6 | 44.2 (-0.4) | 44.7 (+0.1) | 44.4 (-0.2) |
> | | Llama3-8B | 48.6 | 51.0 (+2.4) | 53.5 (+4.9) | 53.4 (+4.8) |
> | CoQA | GPT-4o-mini | 48.4 | 49.4 (+1.0) | 50.8 (+2.4) | 51.0 (+2.6) |
> | | Qwen3-8B | 63.6 | 66.4 (+2.8) | 68.0 (+4.4) | 68.2 (+4.6) |
> | | Llama3-8B | 64.8 | 64.2 (-0.6) | 66.4 (+1.6) | 65.8 (+1.0) |
>
> ---
>
> >**R8 | During the review, I noticed a counter-intuitive phenomenon: more reliable (higher certainty) samples receive higher scores, which is contrary to the general standard for uncertainty scores.**
>
> That is indeed the case. We will unify the notation to **SymmetryScore** (higher→more reliable), explain this in Sections 2.2 and 4.2, and explicitly state in the figure captions that "higher = better" to avoid conflict with the convention that "higher UQ score = more uncertain."
>
> ---
>
> >**R9 | Figure 3's SoRE; the full name of SDLG.**
>
> We will unify "SoRE" in Figure 3 to RES. SDLG is the original abbreviation in the paper, but writing out the full name "Semantically diverse language generation" does appear unusual, so we will remove the full name.

---

> ### Author Response · Authors · 2025-11-12
>
> > **R10 | would -mean($c_{i,fwd}$, $c_{i,bwd}$) be beneficial over the penalized score?**
>
> We did try scoring based on the `contradict` category in early experiments, but the effect was not good. The reason is that when the reasoning and explanation paths are inconsistent in RES, there are two situations:
>
> 1. The situation where the `contradict` score is applicable: There is a **clear logical conflict** between the reasoning and the explanation, such as the case study in Table 5:
>     * Reasoning path: "Starting from May 9th, 2017, adding 40 days brings us to June 17th, 2017."
>     * Explanation path: "Counting 40 days from May 9th leads to June 18th, 2017."
> 2. The situation where the `contradict` score is not applicable: More commonly, the reasoning and explanation are not contradictory, but **lack relevance**, for example:
>     * Reasoning path: "Tom saw a red car pass by, which made him recall his childhood memories."
>     * Explanation path: "Red objects often evoke emotional responses."
>
> The penalized score is able to identify both (1) paths with clear contradictions and (2) paths that are irrelevant, hence it shows better performance.
>
> ---
>
> > **R11 | Regarding the case study, how can the confidence scores in Table 5 be so high?**
>
> We explained in line 465 that the confidence is calculated using the **logits** of the tokens enclosed in the `<final></final>` tag within the answer. The reason for the probability being very close to 1 is also explained there. We believe this is because the question dictates the answer format as MM/DD/YYYY, and this high confidence merely reflects the certainty of the **date format**, not the **logical correctness**.
>
> ---
>
> > **R12 | The performance of RES is weakest on the TriviaQA and CoQA datasets. Do you have any insights into why the method struggled on those datasets or is it rather that baselines were already tuned well for them?**
>
> We addressed this point in line 337 of Section 5.1: "TriviaQA (without documents) largely probes prior knowledge, so errors stem more from model-level hallucination than the question itself (Ji et al, 2023); CoQA answers are mostly extractive, lacking explicit reasoning structure for symmetry checks." Therefore, we emphasized in the abstract that RES has an advantage in complex reasoning tasks.
>
> Furthermore, our method performing "worst" on TriviaQA and CoQA datasets is only when compared **vertically across other datasets**. When compared **horizontally** with other methods, our method is second only to BSDetector. A large part of the reason is that BSDetector additionally introduces an LLM to score the generated answers through several rounds of multiple-choice evaluation. If we also introduce an LLM for scoring, as shown in the table below, our method surpasses BSDetector:
>
> | Dataset | BSDetector | RES (penalized) | RES (penalized) w/ LLM judge |
> | :--- | ---: | ---: | ---: |
> | TriviaQA | 82.5 | 80.0 | **84.2 (+4.2)** |
> | CoQA | 77.9 | 77.6 | **81.1 (+3.5)** |
>
> ---
>
> > **R13 | Recently, Aichberger et al. (2024) demonstrated that the likelihood of a greedy decoding sequence can serve as an effective measure of uncertainty. Since this method is essentially free, I would like to know how it performs as an uncertainty measure in the tasks considered.**
>
> We tested the NLL method proposed by Aichberger et al. on TriviaQA. Compared to earlier baselines LN-PE and SE, this method shows some performance improvement, but it lacks competitiveness when compared to newer baselines.
>
> | metric | LN-PE | NLL | SE | SAR | BSDetector | INSIDE | RES(min) | RES(mean) | RES(penalized) |
> | :--- | ---: | ---: | ---: | ---: | ---: | ---: | ---: | ---: | ---: |
> | LLM | 55.3 | 60.8 | 57.9 | 65.4 | **77.2** | 67.2 | 69.9 | 73.4 | 74.8 |
> | ROUGE-L | 64.5 | 69.0 | 66.7 | 73.5 | **79.5** | 70.6 | 71.5 | 75.3 | 76.5 |

---

### Official Review · Reviewer_49EA · 2025-10-31

**Soundness:** 2
**Presentation:** 3
**Contribution:** 2
**Rating:** 4
**Confidence:** 4

**Summary:**

This paper tackles the cost and instability of semantic uncertainty estimation in LLMs that rely on many sampled outputs. It proposes Reasoning–Explanation Symmetry (RES): for a question, the model first generates structured reasoning and an answer, then, conditioned on the answer, generates a structured explanation; a bidirectional NLI judge computes mutual entailment across aligned sections to yield a symmetry score as an uncertainty proxy. Experiments demonstrate that RES improves AUROC for uncertainty quantification and boosts best-answer selection.

**Strengths:**

- Firstly, I found the overall presentation, including organization and writing of the paper, making it relatively easy to follow their narrative.
- The authors articulate the core problem they're addressing and the motivation to reduce the cost and instability of multi-sample semantic UQ is straightforward. The methodology is intuitive, leveraging a concise concept of internal consistency that makes sense for identifying specific types of errors.
- The experimental evaluation covers a diverse set of tasks and models, which lends some support to the general applicability of their approach. The efficiency gains reported, particularly the need for only a few samples to achieve their results, are a practical advantage that stands out.

**Weaknesses:**

- My main concern is that RES does not address epistemic uncertainty: answer-conditioned explanations can rationalize confident but incorrect outputs, yielding high symmetry and thus underestimating uncertainty when the model simply lacks knowledge. In other words, RES measures process consistency rather than factuality; however, on knowledge-heavy or retrieval-dependent tasks, RES might fail without a fact-checking component.
- The model suite is too small for a reasoning-centric paper: results are confined to ≤8B open models (Llama and Qwen 8B) and a mini closed model (GPT-4o-mini), with no larger size or modern reasoning specialists (contemporary 30–70B or reasoning-tuned variants like QwQ-32B and Deepseek-R1), limiting external validity.
- Scalability to long chains remains untested, as the NLI judge's length limits and section-wise segmentation are potential bottlenecks. Additionally, there is a lack of sensitivity studies on the impact of reasoning/explanation token length on performance.
- In this paper, RES scores pairwise, section-aligned entailment, and then aggregates (by min/mean/penalized). This symmetry aggregation is coarse and overlooks cross-sectional dependencies, allowing global contradictions or step-level failures to slip through. For example, answer-conditioned explanations tend to rationalize the given answer, so R and E can be mutually entailing even if both are self-consistent but wrong relative to the original premise, which RSE (a per-section average) won't penalize.
- The related work discussion didn't cover recent reasoning or explanations focused on UQ directions [1-5], which are directly relevant to this framing.

[1] Mei et.al, Reasoning about Uncertainty: Do Reasoning Models Know When They Don't Know, arXiv 2025.

[2] Zhang et.al, CoT-UQ: Improving Response-wise Uncertainty Quantification in LLMs with Chain-of-Thought, ACL 2025.

[3] Becker et.al, Cycles of thought: Measuring llm confidence through stable explanations, arXiv 2024.

[4] Da et.al, Understanding the uncertainty of llm explanations: A perspective based on reasoning topology, COLM 2025.

[5] Mo et.al, Tree of uncertain thoughts reasoning for large language models, ICASSP 2024.

**Questions:**

Based on the weaknesses I've outlined, here are my main questions for the authors:

- How do the authors distinguish RES’s signal from epistemic uncertainty? Or can the authors give me some evidence that answer-conditioned rationalization doesn’t yield low-uncertainty scores on “confident but wrong” cases?

- Can the method generalize to larger and reasoning-tuned models (30–70B class and reasoning variants), and is there any evidence of size-sensitivity?

- What happens to RES when reasoning/explanation chains get long—do NLI truncation and section segmentation degrade performance, and can the authors provide token-length sensitivity curves?

---

> ### Author Response · Authors · 2025-11-13
>
> Thank you for your review. We appreciate the time and effort you invested in providing constructive feedback that helps us improve our paper. Below is our response.
>
> > **R1 | RES does not address epistemic uncertainty**
>
> We acknowledge that epistemic uncertainty is one source of uncertainty, and in practical engineering, we could incorporate a fact-checking component to make uncertainty quantification more comprehensive. However, given that the assessment of epistemic uncertainty and **aleatoric uncertainty** are currently two parallel research tracks, with distinctly different research focuses and methodologies, including the evaluation of epistemic uncertainty might make the scope of this work too broad. Therefore, this paper focuses on how to quantify aleatoric uncertainty through process consistency, and the baselines compared are also methods that evaluate aleatoric uncertainty.
>
> From another perspective, RES can help us **reverse-infer the type of uncertainty**. For example, when RES yields a low uncertainty score but the result is actually incorrect, we can infer that the model suffers from **epistemic uncertainty** on this type of problem, and further deduce what kind of training data the model might be lacking.
>
> ---
>
> > **R2 | The model suite is too small for a reasoning-centric paper**
>
> As suggested, we have supplemented the experimental results for a larger model in the revised version. In addition to the original Qwen3-8B, we have added an evaluation of **Qwen3-32B**. The results are summarized below (reporting only RES(penalized)'s AUROC and TOP1-AUC):
>
> | Dataset | AUROC (Qwen3-8B) | AUROC (Qwen3-32B) | TOP1-AUC (Qwen3-8B) | TOP1-AUC (Qwen3-32B) |
> | :--- | ---: | ---: | ---: | ---: |
> | MultiRC | 59.8 | 61.0 | 87.2 | 89.0 |
> | Date Understanding | 66.5 | 67.8 | 82.0 | 86.0 |
> | Multistep Arithmetic | 99.2 | 99.6 | 99.6 | 100.0 |
> | StrategyQA | 60.8 | 60.0 | 78.6 | 80.7 |
> | TriviaQA | 75.6 | 72.2 | 47.1 | 51.2 |
> | CoQA | 75.6 | 80.7 | 68.2 | 74.3 |
>
> The formula and pipeline of RES are inherently decoupled from the parameter scale and architecture of the underlying LLM. Furthermore, we have not observed a method-failure-level scale sensitivity in the 8B to 32B range. Therefore, we believe the existing results support the scalability of RES to stronger base models, rather than suggesting it is only applicable to small models.
>
> ---
>
> > **R3 | Scalability to long chains remains untested**
>
> We thank the reviewer for pointing out that the NLI judge's length limits and section segmentation might affect the scalability of long-chain reasoning. For this reason, we have supplemented a sensitivity experiment on **"reasoning token length"** in the revised version. The results show that performance gradually improves and reaches its optimum when the reasoning length is between 300 and 700 tokens; performance slightly decreases when the length limit is increased further. This suggests that overly short chains lead to the NLI judge lacking sufficient evidence to identify fine-grained contradictions, while overly long chains are prone to introducing redundant noise or triggering truncation, thereby weakening the consistency judgment on critical reasoning steps. Given the current evaluation tasks and model scales, RES's performance with respect to length is relatively robust, but we agree that this remains a direction for further research when addressing longer-chain reasoning.
>
> | Max reasoning length L | RES(penalized) (AUROC) |
> | :--- | ---: |
> | 128 | 62.1 |
> | 256 | 64.0 |
> | 384 | 65.4 |
> | 512 | 65.8 |
> | 768 | **66.3** |
> | 1024 | 65.0 |
>
> ---
>
> > **R4 | This symmetry aggregation is coarse and overlooks cross-sectional dependencies, allowing global contradictions or step-level failures to slip through...**
>
> We acknowledge that the granularity of symmetry aggregation might affect the precision of uncertainty quantification. Our structured prompting decomposes the chain into Premise/Evidence, Reasoning, and Conclusion to ensure generality. In fact, for different tasks, the reasoning process can be further refined and aligned.
>
> However, we believe the example you provided is **not** caused by "coarse symmetry aggregation overlooking cross-sectional dependencies." The model repeatedly generating consistent, incorrect answers is a problem stemming from **epistemic uncertainty** (as mentioned in R1). This is caused by the model's knowledge boundary and cannot be resolved by merely considering dependencies between sections. In other words, RES evaluates aleatoric uncertainty, and it is reasonable not to penalize such answers because while the aleatoric uncertainty of the answer and its correctness are correlated, they are not causal.
>
> ---
>
> > **R5 | The related work discussion didn't cover recent reasoning or explanations focused on UQ directions [1-5]**
>
> Thank you for your suggestion. We will supplement the related work section with the literature you mentioned.

---

> ### Author Response · Authors · 2025-11-13
>
> > **Q1 | How do the authors distinguish RES’s signal from epistemic uncertainty? Or can the authors give me some evidence that answer-conditioned rationalization doesn’t yield low-uncertainty scores on “confident but wrong” cases?**
>
> As stated in R1 and R4, the RES signal measures aleatoric uncertainty, which is a parallel research direction to epistemic uncertainty. Our understanding is that "confident but wrong" answers are caused by epistemic uncertainty and can only be evaluated through an external fact-checking component. This has a fundamentally different research focus and paradigm from aleatoric uncertainty, which is assessed through the model's own output or other signals. Therefore, this question is not the focus of this paper's research.
>
> ---
>
> > **Q2 | Can the method generalize to larger and reasoning-tuned models?**
>
> Please see R2.
>
> ---
>
> > **Q3 | What happens to RES when reasoning/explanation chains get long**
>
> Please see R3.

---

### Official Review · Reviewer_2xgL · 2025-10-31

**Soundness:** 2
**Presentation:** 2
**Contribution:** 2
**Rating:** 4
**Confidence:** 4

**Summary:**

The paper introduces RES - Reasoning-Explanation Symmetry as an uncertainty estimation method for natural language generation.
The method roughly consists of generating Evidence and Premises for a Question Answer pair and computing the symmetry score.
The several variations of this score can further be used for ucnertainty estimation.
The authors perform several ablations of their method and compare it to several prominent uncertainty estimation algorithms for autoregressive generation.

**Strengths:**

The method is novel, although highly heuristic.
The evaluation encompasses several multi-step datasets which is a good practice.
Authors perform ablations on the components of their method.

**Weaknesses:**

1. Minor:
    1. Line 038: Semantically Diverse Likelihood Generation (SDLG) - the name is disambiguated incorrectly, it's language, not likelihood.
    2. Evaluation includes TriviaQA and CoQA, I do not see how the method would be applicable to these given their short answer lengths.
    3. Answer length not reported for the evaluation suite.
    4. Reasoning nature of GPT-4o-mini is not treated. The model produces CoT which it does not disclose.
2. Major:
    1. [1][2] say that ROUGE metrics can be quite unreliable for assessing the selective prediction performance of uncertainty estimation algorithms for NLG. Would be nice to show comparison with judge LMs or and average thereof.
    2. PE without length normalization can make a significant difference [3], yet not included in the evaluation.
    3. The proposed pipeline is not well motivated theoretically and has little in the way of integrating it with the theory of preceding uncertainty estimation literature.


### References
1. Ielanskyi, M., Schweighofer, K., Aichberger, L. & Hochreiter, S. Addressing Pitfalls in the Evaluation of Uncertainty Estimation Methods for Natural Language Generation. Preprint at https://doi.org/10.48550/arXiv.2510.02279 (2025).
2. Santilli, A. et al. Revisiting uncertainty quantification evaluation in language models: Spurious interactions with response length bias results. in Proceedings of the 63rd annual meeting of the association for computational linguistics (volume 2: Short papers) (eds Che, W., Nabende, J., Shutova, E. & Pilehvar, M. T.) 743–759 (Association for Computational Linguistics, Vienna, Austria, 2025). doi:10.18653/v1/2025.acl-short.60.
3. Aichberger, L., Schweighofer, K. & Hochreiter, S. Rethinking Uncertainty Estimation in Natural Language Generation. Preprint at https://doi.org/10.48550/arXiv.2412.15176 (2024).

**Questions:**

1. Why does this method work? What theoretical uncertainty quantity if your method closest to?
2. The previous question but on a more applied level: could you compute correlation plots between the RES and other uncertainty estimates for each dataset, could give some interesting insights.
3. How does your method fit in the framework of aleatoric and epistemic uncertainty?
4. What is the computational overhead of the additional generations (e.g. Premises and Evidence)?

---

### Note · Authors · 2025-11-14

**Comment:**

Thank you to all the reviewers for their time and effort. We believe that this paper requires a redesign of the experimental section in order to support our arguments, and therefore we have decided to withdraw the manuscript.

**Withdrawal Confirmation:**

I have read and agree with the venue's withdrawal policy on behalf of myself and my co-authors.